# Gas Exchange in Patients with Pulmonary Tuberculosis: Relationships with Pulmonary Poorly Communicating Fraction and Alveolar Volume

Larisa D. Kiryukhina [1,*], Elena V. Kokorina [1], Pavel V. Gavrilov [1,2], Nina V. Denisova [1], Liudmila I. Archakova [1,2] and Petr K. Yablonskiy [1,2]

1   St. Petersburg Scientific Research Institute of Phthisiopulmonology, 2-4 Ligovsky Avenue, St. Petersburg 191036, Russia
2   Medical Faculty, St. Petersburg State University, 7-9 Universitetskaya Embankment, St. Petersburg 199034, Russia
*   Correspondence: kiryuhina_larisa@mail.ru; Tel.: +7-921-921-56-80

**Abstract:** Tuberculosis-related lung damage is very different. Lung ventilation disorders have been studied in patients with pulmonary tuberculosis (TB) during the active process and after treatment, but the main causes of gas exchange changes have not been sufficiently studied. Investigation of diffusing lung capacity in combination with bodyplethysmography is useful for the interpretation of pulmonary gas exchange disorders. The aim was to determine the relationship of gas exchange with the value of alveolar volume (VA) and pulmonary poorly communicating fraction (PCF) in patients with pulmonary TB. A total of 292 patients (117/175 M/W) with verified pulmonary TB with smoking age less than 10 packs-years underwent spirometry, bodyplethysmography, and $D_{LCO}$ by the single-breath method. PCF was estimated calculating the difference between total lung capacity (TLC) and VA (% TLC). Patients with low $D_{LCO}$ had statistically significantly lower spirometric values (FVC, $FEV_1$, $FEV_1/FVC$, MMEF), lower TLC, higher airway resistance, RV/TLC, air-trapping volume, and PCF. The patients with low level of $D_{LCO}$ were divided into four groups depending on level VA and PCF. In most patients with infiltrative tuberculosis (50%), the leading syndrome of the $D_{LCO}$ decrease was alveolar-capillary damage. In patients with tuberculomas, the syndromes of alveolar capillary damage and pulmonary ventilation inhomogeneity were with the same frequency (43%). In patients with disseminated tuberculosis, the most frequent syndrome of the $D_{LCO}$ decrease was pulmonary ventilation inhomogeneity (33%), then alveolar-capillary damage (29%) and mixed (24%). In patients with cavernous tuberculosis, the leading syndrome of the $D_{LCO}$ decrease was mixed (39%), then alveolar capillary damage (25%) and pulmonary ventilation inhomogeneity (23%). The syndrome of gas exchange surface reduction in patients with disseminated and cavernous tuberculosis was less common (14%). In conclusion, an additional evaluation of the combination of PCF and VA increases the amount of clinical information obtained using the diffusion lung capacity measurements, since it allows identifying various syndromes of gas exchange impairment. The leading causes of diffusing capacity impairment vary by different types of pulmonary TB.

**Keywords:** pulmonary tuberculosis; diffusion lung capacity; pulmonary gas exchange; bodyplethysmography; poorly communicating fraction

## 1. Introduction

The diffusing capacity of the lung (transfer factor) is tested for measuring of gas transfer from alveoli to pulmonary capillary blood, the results depend on the lung structural and functional properties [1]. The most common methodology for diffusion capacity testing is the measurement of carbon monoxide (CO) uptake ($D_{LCO}$). $D_{LCO}$ is the product of two measurements: the rate constant for CO uptake from alveolar gas ($k_{CO}$ (minute$^{-1}$)) and the "accessible" alveolar volume (VA). The $k_{CO}$ expressed per mm Hg alveolar dry gas

pressure (Pb) ($k_{CO}$/Pb-47) equals $D_{LCO}$ divided by VA ($D_{LCO}$/VA, also called transfer coefficient $K_{CO}$) [2]. When assessing changes in the diffusion capacity, it is necessary to rely not only on the value of $D_{LCO}$, but also on the value of the alveolar volume and on the carbon monoxide transfer coefficient. One of the misconceptions is that $K_{CO}$ is the value of $D_{LCO}$ "adjusted" for lung volume [3]. The volume of pulmonary capillary blood remains relatively constant as lung volume decreases. $D_{LCO}$ decreases linearly with decreasing alveolar volume, while $D_{LCO}$/VA increases non-linearly [4]. The evaluation of $K_{CO}$ helps in the diagnosis of various pathological processes at the same $D_{LCO}$ value [2,3]. Gas exchange abnormality may occur with various combinations of transfer coefficient and alveolar volume.

Decreased $K_{CO}$ occurs in alveolar-capillary damage, microvascular pathology, or anemia. The alveolar volume is useful as a characteristic of the gas exchange surface. Decreased VA occurs in restrictive lung diseases, alveolar damage, or loss, or small airways obstruction [2,5]. The ventilation maldistribution will cause an underestimation of VA and at the same time an overestimation of $K_{CO}$ [6]. Adequate inhaled air distribution is important for effective breathing and gas exchange [7]. The valuable information regarding inspired gas distribution abnormalities and trapped gas volume may be received with measurement of the "poorly communicating fraction" of total lung capacity (PCF) [8,9].

Tuberculosis causes different lesions of lung parenchyma and bronchi and bronchioles, leading to gas exchange impairment [10–12]. We hypothesized that diffusing capacity impairment in patients with different tuberculosis disorders may be associated with different leading causes. For better understanding the clinical implications of $D_{LCO}$ in patients with pulmonary tuberculosis (TB), we studied the relationship of gas exchange with value of alveolar volume and pulmonary poorly communicating fraction as a marker of pulmonary ventilation inhomogeneity.

## 2. Materials and Methods

### 2.1. Subjects

This study involved a retrospective analysis of data collected between 2018 and 2021 at the Respiratory Investigation Unit, Thoracic Center of St. Petersburg Scientific Research Institute of Phthisiopulmonology (Russia). All patients had a verified diagnosis of pulmonary TB. The St. Petersburg Scientific Research Institute of Phthisiopulmonology Ethics Board approved the use of the anonymous data. Inclusion criteria: included adult men and women, availability of both TLC by bodyplethysmography and VA from $D_{LCO}$ measurements, smoking age ≤10 years. We excluded patients who met the following criteria: patients <18 years old; patients without laboratory-confirmed TB; women who were pregnant at the time of the hospitalization; orthopedic, neuromuscular, cardiac, or metabolic conditions preventing the patient from safely undertaking pulmonary function tests; the presence of COPD, asthma, or any lung disease other than TB. According to these criteria, 292 patients were selected. Patients had taken part in ethically approved research studies in which pulmonary function tests were performed as part of the study entry assessment. Written informed consent was obtained from every patient.

### 2.2. Study Design

All patients underwent pulmonary function testing (PFT) including spirometry, bodyplethysmography, investigation of diffusing lung capacity and chest computed tomography (CT).

### 2.3. Pulmonary Function Measurements

All patients underwent PFT using MasterScreen Body Diffusion (VIASYS Healthcare, Höchberg, Germany). Forced vital capacity (FVC), forced expiratory volume in 1 s ($FEV_1$), maximal mid-expiratory flow (MMEF), total airways resistance (Rtot), total lung capacity (TLC), residual volume (RV), and ratio of RV to TLC (RV/TLC) were recorded. $D_{LCO}$ and $K_{CO}$ were measured by the single-breath technique using measuring gas with 0.26% CO,

9% helium, 19% oxygen, rest—nitrogen. $D_{LCO}$ was corrected for hemoglobin. The tests were performed and results interpreted using the American Thoracic Society/European Respiratory Society guidelines [3,13–15]. Predicted values were determined using the formulae of European Coal and Steel Community 1993 [16]. Abnormal values were considered the values of ventilation parameters and gas exchange outside from the 95% confidence interval (outside lower and upper limits of normal). VA was considered normal $\leq$80% predicted.

The noncommunicating gas ("air-trapping volume") was determined as difference between TLC (plethysmograph) and TLC (helium) [17].

The PCF was estimated calculating the difference between total lung capacity and alveolar volume (1—(VA/TLC) (%). A PCF value $\leq$15% was considered as normal, $\leq$23% as "mild" pulmonary ventilation inhomogeneity, 24–33% as "moderate", and $\geq$34% as "extensive" [8].

### 2.4. Image Analysis

All patients (n = 292) underwent a chest CT with a slice thickness of 1 mm and standard scanning parameters on TOSHIBA tomographs. The analysis of the size of tuberculosis foci (volume of maximal focus, total volume of foci, destruction zone volume) was carried out on 165 patients using the Nodule Analysis application software package (TOSHIBA).

### 2.5. Statistical Analysis

Values are reported as median (interquartile range) unless otherwise specified. Comparisons across subgroups were performed using the Mann–Whitney U test and analysis of variance (ANOVA) with post hoc testing of significant variables carried out using *t* tests with Bonferroni adjustment for multiple comparisons. Yates-corrected Chi-square analysis tested the association between categorical variables. A *p* value of < 0.05 was considered significant in Mann–Whitney U test and *p* < 0.008 for multiple comparisons.

### 3. Results

Our group of participants consisted of 292 patients (men 40%) from 18 to 71 years of age. We classified them into two groups based on $D_{LCO}$ measurement results (Table 1). According to the table, we can observe that the patients in the groups with reduced and normal $D_{LCO}$ did not differ in age, gender, or body mass index. The number of smoking patients did not differ in both groups either. In both groups, smokers had little smoking experience. Despite the short smoking experience, the number of pack-years turned out to be significantly higher in the group with reduced diffusion lung capacity.

**Table 1.** Characteristics of patients with pulmonary tuberculosis depending on gas exchange (n = 292).

| Characteristics | $D_{LCO} \geq$ LLN n = 96 | $D_{LCO} <$ LLN n = 196 | *p* |
|---|---|---|---|
| Male/female gender | 37/59 | 80/116 | ns |
| Age years | 30 (25–38) | 31 (27–40) | ns |
| Body mass index kg·m$^{-2}$ | 21.4 (19.5–24.2) | 20.9 (18.9–23.5) | ns |
| Smoking history no/yes, n (%) | 48 (50)/48 (50) | 80 (41)/116 (59) | ns |
| Pack-years | 0.3 (0–4.8) | 2 (0–6.0) | 0.029 |
| Forms of Pulmonary Tuberculosis | | | |
| Infiltrative | 20 (21) | 16 (8) | 0.004 |
| Tuberculoma | 32 (33) | 30 (15) | 0.001 |
| Disseminated | 7 (7) | 21 (11) | ns |
| Cavernous | 37 (39) | 129 (66) | <0.001 |
| Number of foci | 2.0 (1.5–3.0) | 3.0 (2.0–3.0) | <0.001 |

**Table 1.** *Cont.*

| Characteristics | $D_{LCO} \geq$ LLN n = 96 | $D_{LCO} <$ LLN n = 196 | *p* |
|---|---|---|---|
| Volume of maximal focus mm$^3$ (n = 165) | 7550 (5750–13,300) | 13,500 (7800–38,400) | <0.001 |
| Total volume of foci mm$^3$ (n = 165) | 13,900 (7600–24,200) | 31,700 (16,700–109,600) | <0.001 |
| Destruction zone volume mm$^3$ (n = 165) | 425 (0–10,400) | 8850 (500–45,600) | <0.001 |
| FVC% predicted | 107.7 (97.1–115.2) | 88.6 (73.6–102.9) | <0.001 |
| FEV$_1$% predicted pre | 100.3 (92.1–113.5) | 79.7 (63.8–98.5) | <0.001 |
| FEV$_1$% predicted post | 104.5 (97.1–115.8) | 86.7 (65.8–102.2) | <0.001 |
| FEV$_1$/FVC% pre | 81.0 (75.9–88.2) | 77.8 (72.9–84.1) | <0.001 |
| FEV$_1$/FVC% post | 84.6 (80.1–90.2) | 81.1 (75.7–86.4) | 0.001 |
| MMEF% predicted | 77.9 (58.1–98.5) | 52.2 (32.3–76.1) | <0.001 |
| Rtot% predicted | 76.2 (60.2–104.1) | 99.2 (72.9–141.9) | <0.001 |
| TLC% predicted | 112.3 (103.9–123.0) | 102.1 (88.4–112.7) | <0.001 |
| RV% predicted | 133.2 (114.4–150.9) | 129.7 (109.5–152.0) | ns |
| RV/TLC% predicted | 114.0 (105.0–127.9) | 124.5 (109.2–143.0) | <0.001 |
| $D_{LCO}$% predicted | 88.4 (82.9–95.7) | 67.7 (58.3–73.0) | <0.001 |
| $K_{CO}$% predicted | 87.7 (79.8–96.5) | 77.2 (70.8–84.7) | <0.001 |
| VA% predicted | 103.5 (96.1–109.3) | 85.2 (73.3–97.9) | <0.001 |
| VA < 80% predicted | 1 (1) | 83 (42) | <0.001 |
| Air-trapping volume L | 0.33 (0.01–0.65) | 0.55 (0.30–0.92) | <0.001 |
| PCF% TLC | 12.0 (6.1–16.2) | 16.2 (12.2–22.9) | <0.001 |
| PCF > 15% | 28 (29) | 113 (58) | <0.001 |

Data are presented as n, median (interquartile range) or n (%), unless otherwise stated. LLN: lower limit of normal; FVC: forced vital capacity; FEV$_1$: forced expiratory volume in 1 s; MMEF: maximal mid-expiratory flow; Rtot: total airways resistance; TLC—total lung capacity; RV—residual volume; $D_{LCO}$: diffusing capacity of the lung for carbon monoxide; VA—alveolar volume; $K_{CO}$—transfer coefficient of the lung for carbon monoxide; PCF—poorly communicating fraction.

Infiltrative tuberculosis and tuberculomas were more often detected in patients with normal $D_{LCO}$, cavernous tuberculosis was more often detected in the group with reduced $D_{LCO}$, disseminated tuberculosis was found with the same frequency in both groups. In the group with abnormal $D_{LCO}$, there were significantly more tuberculosis foci, higher total volume of foci, and total volume of destruction zone.

Patients with low $D_{LCO}$ had significantly lower values of spirometric parameters (FVC, FEV$_1$, FEV$_1$/FVC, MMEF), lower TLC, higher airway resistance, RV/TLC, air-trapping volume, and PCF.

In the group with normal DLCO, compared with the group with abnormal DLCO, there were fewer patients with abnormal VA levels (1% vs. 42%, *p* < 0.001) and abnormal PCF levels (29% vs. 58%, *p* < 0.001). In the group with normal $D_{LCO}$ and a high level of PCF, all patients (100%) had "mild" PCF. In the group with low $D_{LCO}$ and high level of PCF, 57.5% had "mild" PCF, 42.5%—"moderate" and "extensive".

In pulmonary TB patients with the same volume of tuberculous lesion, the severity of pulmonary gas exchange disorders differed. Among these patients, groups were identified depending on the alveolar volume and pulmonary poorly communicating fraction values.

We compared the results of patients with low level of $D_{LCO}$ depending on level VA and PCF. Patients were divided to four groups: 1—VA $\geq$ 80% predicted, PCF < 15%; 2—VA $\geq$ 80% predicted, PCF > 15%; 3—VA < 80% predicted, PCF < 15%; 4—VA < 80% predicted, PCF > 15% (Table 2).

**Table 2.** Characteristics of patients with pulmonary tuberculosis with low level diffusion lung capacity depending on alveolar volume and pulmonary poorly communicating fraction (n = 196).

| Characteristics | VA ≥ 80% Predicted | | VA < 80% Predicted | | $p$ |
|---|---|---|---|---|---|
| | **PCF < 15%** **n = 59** | **PCF > 15%** **n = 54** | **PCF < 15%** **n = 24** | **PCF > 15%** **n = 59** | |
| | **1** | **2** | **3** | **4** | |
| Forms of Pulmonary Tuberculosis | | | | | |
| Infiltrative | 8 (50) | 5 (31) | 2 (13) | 1 (6) | |
| Tuberculoma | 13 (43.3) | 13 (43.3) | 1 (3.3) | 3 (10) | |
| Disseminated | 6 (28.6) | 7 (33.3) | 3 (14.3) | 5 (23.8) | |
| Cavernous | 32 (24.8) | 29 (22.5) | 18 (14.0) | 50 (38.7) | |
| Total volume of foci mm$^3$ (n = 113) | 27,400 (14,100–49,600) | 19,600 (14,000–46,600) | 30,650 (15,700–98,400) | 104,050 (30,400–284,250) | $p_{1-2} = 0.643$ $p_{1-3} = 0.741$ $p_{1-4} < 0.001$ $p_{2-3} = 0.603$ $p_{1-4} < 0.001$ $p_{3-4} = 0.044$ |
| Destruction zone volume mm$^3$ (n = 113) | 5700 (2–13,200) | 4650 (0–26,400) | 4700 (2900–40,300) | 36,650 (7150–159,225) | $p_{1-2} = 1.000$ $p_{1-3} = 0.209$ $p_{1-4} < 0.001$ $p_{2-3} = 0.257$ $p_{2-4} < 0.001$ $p_{3-4} = 0.084$ |
| FVC% predicted | 102.5 (93.3–112.4) | 98.2 (90.7–108.1) | 74.2 (64.9–80.4) | 68.6 (58.5–77.9) | $p_{1-2} = 0.157$ $p_{1-3} < 0.001$ $p_{1-4} < 0.001$ $p_{2-3} < 0.001$ $p_{2-4} < 0.001$ $p_{3-4} = 0.116$ |
| FEV$_1$% predicted pre | 98.6 (90.4–105.7) | 91.9 (78,9–102.2) | 71.6 (60.5–76.1) | 59.8 (46.8–69.7) | $p_{1-2} = 0.032$ $p_{1-3} < 0.001$ $p_{1-4} < 0.001$ $p_{2-3} < 0.001$ $p_{2-4} < 0.001$ $p_{3-4} = 0.003$ |
| FEV$_1$/FVC% pre | 81.0 (73.9–85.1) | 77.8 (71.5–84.2) | 82.7 (76.7–86.8) | 73.8 (67.5–77.9) | $p_{1-2} = 0.033$ $p_{1-3} = 0.700$ $p_{1-4} < 0.001$ $p_{2-3} = 0.079$ $p_{2-4} = 0.004$ $p_{3-4} < 0.001$ |
| MMEF% predicted | 75.7 (50.3–96.1) | 63.4 (46.1–78.2) | 54.4 (34.4–64.7) | 23.9 (20.6–42.4) | $p_{1-2} = 0.018$ $p_{1-3} < 0.001$ $p_{1-4} < 0.001$ $p_{2-3} = 0.132$ $p_{2-4} < 0.001$ $p_{3-4} < 0.001$ |
| Rtot% predicted | 75.9 (61.1–96.0) | 93.7 (74.4–124.5) | 112.8 (64.9–140.4) | 154.6 (103.9–204.9) | $p_{1-2} = 0.003$ $p_{1-3} = 0.009$ $p_{1-4} < 0.001$ $p_{2-3} = 0.490$ $p_{2-4} < 0.001$ $p_{3-4} = 0.004$ |

**Table 2.** *Cont.*

| Characteristics | VA ≥ 80% Predicted | | VA < 80% Predicted | | *p* |
|---|---|---|---|---|---|
| | PCF < 15%<br>n = 59 | PCF > 15%<br>n = 54 | PCF < 15%<br>n = 24 | PCF > 15%<br>n = 59 | |
| | 1 | 2 | 3 | 4 | |
| TLC% predicted | 106.3<br>(100.1–115.4) | 113.7<br>(106.3–118.4) | 82.8<br>(72.7–86.3) | 89.4<br>(81.6–98.8) | $p_{1-2} = 0.004$<br>$p_{1-3} < 0.001$<br>$p_{1-4} < 0.001$<br>$p_{2-3} < 0.001$<br>$p_{2-4} < 0.001$<br>$p_{3-4} = 0.001$ |
| RV% predicted | 124.0<br>(109–137.4) | 145.3<br>(124.7–161.2) | 99.3<br>(84.5–123.1) | 135.7<br>(108.6–155.6) | $p_{1-2} < 0.001$<br>$p_{1-3} < 0.001$<br>$p_{1-4} = 0.049$<br>$p_{2-3} < 0.001$<br>$p_{2-4} = 0.047$<br>$p_{3-4} < 0.001$ |
| RV/TLC%<br>predicted | 109.7<br>(97.2–125.2) | 124.5<br>(112.3–140.9) | 122.7<br>(105.0–139.2) | 142.1<br>(125.9–170.5) | $p_{1-2} < 0.001$<br>$p_{1-3} = 0.061$<br>$p_{1-4} < 0.001$<br>$p_{2-3} = 0.417$<br>$p_{2-4} < 0.001$<br>$p_{3-4} < 0.001$ |
| $D_{LCO}$% predicted | 71.9<br>(68.2–75.1) | 69.7<br>(65.6–75.5) | 61.5<br>(55.9–69.3) | 57.6<br>(48.6–64.7) | $p_{1-2} = 0.321$<br>$p_{1-3} < 0.001$<br>$p_{1-4} < 0.001$<br>$p_{2-3} < 0.001$<br>$p_{2-4} < 0.001$<br>$p_{3-4} = 0.015$ |
| $D_{LCO}$ < 60%<br>predicted | 2 (3) | 4 (7) | 9 (38) | 39 (66) | |
| VA% predicted | 98.8<br>(91.5–105.2) | 91.8<br>(86.9–99.0) | 76.8<br>(71.9–78.5) | 69.3<br>(60.9–74.6) | $p_{1-2} = 0.002$<br>$p_{1-3} < 0.001$<br>$p_{1-4} < 0.001$<br>$p_{2-3} < 0.001$<br>$p_{2-4} < 0.001$<br>$p_{3-4} < 0.001$ |
| $K_{CO}$% predicted | 72.4<br>(68.2–78.7) | 76.4<br>(71.6–81.4) | 83.9<br>(76.4–90.2) | 81.8<br>(74.2–93.2) | $p_{1-2} = 0.131$<br>$p_{1-3} = 0.002$<br>$p_{1-4} = 0.01$<br>$p_{2-3} = 0.018$<br>$p_{2-4} = 0.106$<br>$p_{3-4} = 0.485$ |
| Hemoglobin<br>g·100 mL$^{-1}$ | 13.5 (12.6–14.7) | 13.4 (12.5–14.4) | 12.8 (11.9–14.1) | 12.5 (11.1–13.6) | $p_{1-2} = 0.834$<br>$p_{1-3} = 0.064$<br>$p_{1-4} = 0.006$<br>$p_{2-3} = 0.087$<br>$p_{2-4} = 0.010$<br>$p_{3-4} = 0.581$ |
| Air-trapping<br>volume L | 0.34<br>(0.14–0.50) | 0.77<br>(0.61–1.13) | 0.19<br>(0.07–0.34) | 0.9<br>(0.57–1.3) | $p_{1-2} < 0.001$<br>$p_{1-3} = 0.035$<br>$p_{1-4} < 0.001$<br>$p_{2-3} < 0.001$<br>$p_{2-4} = 0.355$<br>$p_{3-4} < 0.001$ |

**Table 2.** *Cont.*

| Characteristics | VA ≥ 80% Predicted | | VA < 80% Predicted | | *p* |
|---|---|---|---|---|---|
| | PCF < 15% n = 59 | PCF > 15% n = 54 | PCF < 15% n = 24 | PCF > 15% n = 59 | |
| | 1 | 2 | 3 | 4 | |
| PCF% TLC | 11.3 (7.5–13.4) | 19.0 (16.7–22.6) | 10.9 (7.4–12.7) | 25.3 (20.3–30.3) | $p_{1-2} < 0.001$ $p_{1-3} = 0.619$ $p_{1-4} < 0.001$ $p_{2-3} < 0.001$ $p_{2-4} < 0.001$ $p_{3-4} < 0.001$ |

Data are presented as n, median (interquartile range) or n (%), unless otherwise stated.

In the 1st group with normal levels of VA and PCF both, the median value ventilation parameters were within normal limits. There were 41 patients (69%) with normal ventilation, 16 (27%) with mild obstruction, 1 patient with mild restriction (2%), and 1 patient with mixed disorders (2%). The decrease in $D_{LCO}$ was mild and accompanied by $K_{CO}$ decline. There were no signs of anemia, so the main cause for $D_{LCO}$ reducing was alveolar-capillary barrier damage.

In the 2nd group with normal VA and high PCF, a mild decrease in the median value of the MMEF was observed, which shows an obstruction of the distal airways. There was also an increase in the residual lung volume and the air-trapping volume. There were 26 patients (48%) with normal ventilation and 28 (52%) with mild to moderate obstruction. The decrease in $D_{LCO}$ was mild. The total volume of foci and destruction zone volume did not differ from group 1, but the parameters of airway flow ($FEV_1$, $FEV_1/FVC$, MMEF) were significantly lower, and airway resistance, residual lung volume, and the air-trapping volume were significantly higher. Therefore, the leading reason for the $D_{LCO}$ decrease, in addition to alveolar-capillary barrier damage, was the pulmonary ventilation inhomogeneity.

In the 3rd group with decreased VA and normal PCF, there was a decrease in the lung volumes (FVC, $FEV_1$, TLC). Most of the patients in this group had restrictive (58%) or mixed (21%) ventilation disorders. There were only 3 patients (13%) with normal ventilation and 2 (8%) with mild obstruction. The median value of $D_{LCO}$ was significantly lower than in group 1 and 2. One third of pulmonary TB patients in this group (34%) had moderate decrease of $D_{LCO}$ (40–60% predicted) and in 4%, $D_{LCO}$ was decreased severely (less 40% predicted). The total volume of foci and the volume of destruction zone did not differ from groups 1 and 2. The alveolar volume decrease was accompanied by the proportional decrease in TLC, FVC, $FEV_1$; $K_{CO}$ and the air-trapping volume were within normal limits. Thus, we concluded that the main cause of gas exchange abnormality in this group was gas exchange surface reduction.

In the 4th group with decreased VA and high PCF, obstructive (51%) and mixed (32%) disorders prevailed, restrictive disorders were less common (12%), and normal ventilation was in single cases (5%). The total volume of foci and destruction zone volume were significantly larger than in groups 1–3. The ventilation disorders were more significant compared with groups 1–3. The lung volume and capacity changes showed decreasing of FVC and TLC and increasing of RV and RV/TLC; impaired airway flow was presented as decreased $FEV_1$, MMEF, and $FEV_1/FVC\%$ and increased Rtot. The pulmonary gas exchange dysfunction was manifested as considerable $D_{LCO}$ reduction: 58% of patients had moderate and 8% severe decrease of $D_{LCO}$. The median $D_{LCO}$ value was significantly lower compared to groups 1–3. Thus, the most severe lesion of pulmonary gas exchange in pulmonary tuberculosis is associated with the summation of several main causes. There was a loss of lung volume with a loss of the alveolar-capillary structure, which led to a decrease in the gas exchange surface and there was a high pulmonary poorly communicating fraction, too.

This made it possible to distinguish this syndrome as a mixed variant of pulmonary gas exchange disorders.

Analysis of the distribution of syndromes of pulmonary gas exchange impairment in different clinical forms of pulmonary TB showed that in patients with infiltrative tuberculosis, the alveolar capillary damage syndrome was the main cause of $D_{LCO}$ reduction (50%). The syndrome of pulmonary ventilation inhomogeneity was less common (31%). In patients with tuberculomas, syndromes of alveolar-capillary barrier damage and pulmonary ventilation inhomogeneity were with the same frequency (43%). The other syndromes in these pulmonary TB forms were in single cases.

In patients with disseminated tuberculosis, the most frequent syndrome of the $D_{LCO}$ decrease was pulmonary ventilation inhomogeneity (33%), then alveolar-capillary barrier damage (29%) and mixed (24%). Gas exchange surface reduction syndrome was less common (14%).

It was found that the leading syndrome of the $D_{LCO}$ decrease in patients with cavernous tuberculosis was mixed (39%), followed by alveolar-capillary barrier damage (25%), pulmonary ventilation inhomogeneity (23%), and gas exchange surface reduction (14%).

## 4. Discussion

Thus, more than half of the observed pulmonary TB patients (67%) had pulmonary gas exchange disorders. We included patients with short smoking experience in the study to assess the impact of the tuberculosis process itself on the state of pulmonary gas exchange. The number of smoking and non-smoking patients in the groups with normal diffusion capacity and reduced diffusion level did not differ. However, in the group with reduced diffusion, this seemingly insignificant smoking experience was significantly higher. This is probably since with pulmonary tuberculosis, even a short smoking experience significantly enhances the process of lung destruction. This observation requires further study.

The diffusion lung capacity measurement is widely used in various respiratory diseases. However, in clinical practice, there is no conviction in the informativeness of the additional parameters that we receive when determining $D_{LCO}$—alveolar volume and transfer-coefficient. The informativeness and the need to use the transfer-coefficient in interpreting the $D_{LCO}$ results caused an especially active discussion [2,6,18]. Interesting data were obtained by M. Kameneva, who proposed to identify the causes of pulmonary gas exchange abnormalities in patients with interstitial lung diseases (ILD) by comparing the value of the alveolar volume and the air-trapping volume [19]. In this study, some doubts about the diagnostic significance of $K_{CO}$ in ILD patients were expressed because in 30% of ILD patients with reduced $D_{LCO}$ and bilateral interstitial changes in the lungs $K_{CO}$ remained normal.

Currently, in the latest recommendations on the interpretation of functional tests, $K_{CO}$ is included in the $D_{LCO}$ data interpretation algorithm. It is also mentioned that it is useful to compare VA to TLC measured by body plethysmography to determine whether the incorrect distribution of the tested gas may contribute to decreasing in $D_{LCO}$ [4].

Tuberculosis causes changes in the structure of the lungs and bronchi, which differ in prevalence and morphological characteristics. We have suggested that in various clinical forms of pulmonary tuberculosis, various leading causes of pulmonary gas exchange disorders are possible. Based on a comparison of the alveolar volume, the "poorly communicating fraction" of the total lung capacity, and taking into account the $K_{CO}$ value, we identified the leading syndromes of pulmonary gas exchange disorders: damage to the alveolar-capillary barrier, inhomogeneity of lung ventilation, reduction of the gas exchange surface, and mixed.

Diffusing capacity measurement is the often-ignored lung function test in TB patients; most of the research dates back to the 1960s–1980s. M.H. Williams and co-authors showed that the relationship between the diffusion disorder and the degree of radiographic anomaly was good, while the relationship between the decrease in vital capacity and the degree of radiographic anomaly was poor. They concluded that the diffusion capacity may

be a more sensitive and accurate indicator of the degree of pathological tuberculosis damage than a chest X-ray [20]. In the study by F. Dietiker and coauthors, it was found that the diffusion capacity in pulmonary TB patients correlates well with lung volume. However, they concluded that routine determinations of diffusion capacity in patients with pulmonary tuberculosis add little useful information to the ventilation measurements, and that "alveolar capillary blockade" is not characteristic of any of the common forms of this disease [21].

The leading factors of decreasing in $D_{LCO}$ in TB patients were determined in the study by V. Nefedov and co-authors, in which in addition to $D_{LCO}$ and $K_{CO}$, membrane conductance Dm and effective capillary blood volume Vc were compared with various forms of pulmonary TB [22]. In this study, the authors concluded that the leading factor in the $D_{LCO}$ reduction in patients with disseminated and cavernous tuberculosis was a decrease in the respiratory surface of the lungs because of a decrease in the effective alveolar volume; the leading factor in infiltrative tuberculosis was a decrease in the permeability of the alveolocapillary membrane. Our conclusions on the leading cause of the DLCO decrease in patients with infiltrative tuberculosis are similar to those made by Nefedov and co-authors but differ in patients with disseminated and cavernous tuberculosis. This is probably because Nefedov's study did not consider the gas distribution abnormalities and air-trapping volume.

The proposed concept of the interpretation of the diffusion capacity measurement in combination with bodyplethysmography is useful for understanding the causes of gas exchange abnormality and identifying individual features of the lung diseases. We suppose the findings can also help the management of TB patients.

For patients with pulmonary ventilation heterogeneity, it may be useful to add combinations of bronchodilators that reach the distal airways. Bronchial obstruction has a negative impact on the tuberculosis course, the effectiveness of chemotherapy and the quality of life of patients with pulmonary tuberculosis [23–25]. The use of various bronchodilators in patients with pulmonary tuberculosis with a good clinical effect and a positive effect on the effectiveness of etiotropic therapy was described. The positive effect of broncholytic therapy on the effectiveness of TB treatment has been proven by accelerating the timing of abacillation, closing the decay cavities, and improving the quality of life [26,27]. Unfortunately, these studies did not look at pulmonary gas exchange. In addition, bronchodilators are prescribed in the presence of bronchial obstruction according to spirometry. In our study, some patients (48%) did not have lung ventilation disorders by spirometry, but they had a decrease in the diffusion lung capacity due to an inhomogeneity of pulmonary ventilation.

For patients with alveolar capillary barrier damage, improving microcirculation in the lungs has potential benefits. For example, pentoxifylline improves microcirculation and rheological properties of blood and has anti-inflammatory and immunomodulatory effects. The effectiveness of pentoxifylline has been identified in the treatment of various lung diseases, including infectious causes. This drug has also been shown to reduce pulmonary fibrosis in patients with COVID-19 [28,29].

Thus, improving microcirculation and reducing the burden of hyperinflation will be useful for restoring the homogeneity of the ventilation–perfusion relationships in the lungs and, finally, for improving gas exchange in TB patients. These assumptions have not been proven in the present study and require further studies.

## 5. Conclusions

Gas exchange abnormalities may occur with various combinations not only of $K_{CO}$ and VA. An additional evaluation of the combination of PCF and VA increases the amount of clinical information obtained using the diffusion lung capacity measurements, since it allows identifying various syndromes of gas exchange impairment. The leading causes of diffusing capacity impairment vary by different types of pulmonary TB.

**Author Contributions:** Conceptualization, P.K.Y. and L.D.K.; methodology, L.D.K.; formal analysis, L.D.K., E.V.K., P.V.G., N.V.D., L.I.A. and P.K.Y.; writing—original draft preparation, L.D.K.; writing—review and editing, L.D.K., E.V.K., P.V.G., N.V.D., L.I.A. and P.K.Y.; supervision, P.K.Y.; project administration, L.I.A. All authors have read and agreed to the published version of the manuscript.

**Funding:** This research received no external funding.

**Institutional Review Board Statement:** The study was conducted in accordance with the Declaration of Helsinki, and approved by the Independent Ethics Committee of the St Petersburg Scientific Research Institute of Phthisiopulmonology (Protocol No. 96 of 22 March 2023).

**Informed Consent Statement:** Written informed consent was obtained from every patient.

**Data Availability Statement:** The data presented in this study are available on request from the corresponding author. The data are not publicly available due to patient and hospital information privacy.

**Conflicts of Interest:** The authors have no conflict of interest to declare. The authors bear sole responsibility for the content and writing of the paper.

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
