# Peer review of "Gas Exchange in Patients with Pulmonary Tuberculosis: Relationships with Pulmonary Poorly Communicating Fraction and Alveolar Volume"

_2673-527X, doi:10.3390/jor3020011_

Round 1

Reviewer 1 Report

The authors evaluated diffusing capacity impairment in patients with different types of tuberculosis disorders and found that the leading causes of impairment varied by different types of TB. The aim of the study is innovative, and the finding has great scientific merit, but this manuscript needs substantial revision before publication.

Major Concerns

1.       The authors demonstrated that the impairment of diffusion capacity in patients with different types of tuberculosis disorders may have different leading causes. However, how this additional clinical information would help the management of the TB patient was not clearly addressed in the introduction or the discussion. Most of the content in the Discussion section is actually results and should be moved to the Result section. The result section needs to be rewritten. The current version of the results (presented in the Discussion section in the current version) is difficult for the reader to follow.  Although there are a lot of significant findings, the authors need to summarize the important findings in a more structural way. Many of the clinical variable in Table 2 are highly correlated, so the authors may consider focusing on some representative ones.

2.        The entire discussion section needs to be rewritten. As I said, most of the words in the current Discussion section need to be moved to the Result section. Please discuss whether the finding of this manuscript is the same/different from other relevant studies, or/and discuss how the findings can help to manage TB patients, or/and may help predict post-TB lung function impairment… etc.

Minor concerns

1.       Table 1 showed that all patients with Dlco>=LLN were not smokers. However, 88 of 116 patients with DLCO<LLN were smokers. To remove the confounding introduced by tobacco use, the authors may consider repeating the analyses in a subgroup of patients who were not smokers as a sensitive analysis.

2.       If the p-values presented in the tables are after multiple comparison adjustment, please specify it clearly in the tables.

3.       In the analysis section, it showed “The Shapiro–Wilk test for normality was performed’. First, I am not sure why normality tests would be needed for this study given the sample size of this study. Second, please present the test results in the Results section if they were done.

4.       In the analysis section, it showed “tests with Bonferroni adjustment for multiple comparisons’. However, in the two sentences after this sentence, it showed that “A p value of < 0.05 was considered significant in all analyses”. Please confirm that the p-value cutoff was after Bonferroni adjustment. Please also specified how the adjustment were applied.

6.       Some characteristics listed in Table 2 were not even mentioned in the method section. Please add them.

7.       What is the difference between “<” and “<<” in the table?

Many long sentences need “comma”. For example, in the last sentence of intro a “comma” is needed between “tuberculosis (PT)” and “we”.  Please, have the revised version be checked by a native English speaker.

Author Response

Dear colleague,
We are extremely grateful for such a thorough review of our manuscript. We found these comments very important. All comments have been taken into account and corrected. Please see the attachment. We hope that now the article has become better and clearer.
With best regards and great appreciation.

Reviewer 2 Report

Minor Comments

In this research article, the authors investigate the relationship of gas exchange with value of alveolar volume (VA) and pulmonary poorly communicating fraction (PCF) in patients with PT. The manuscript is well written, and the experimental design/data analysis are robust.  I would recommend the following minor comments to the authors.

Point 1: It would be better to rewrite sentences.

1. In the group with decreased VA and normal PCF, the median value of DLCO was mild decreased too, but it was statistically significantly lower than in group 1 and 2.

2.  This combination of functional characteristics made it possible to distinguish the syndrome of the gas ex- change surface reduction as the main cause of gas exchange abnormality in this group.

Point 2: Minor English correction is required in the revised version.

Point 3: It would be nice for the authors to draw a detailed experimental workflow chart. Otherwise, it would be difficult for the reader to capture the overall picture of the study. Overall, I could not really fault the experiments or the interpretation.

Good Luck

Point 2: Minor English correction is required in the revised version.

Author Response

Dear colleague,
I am extremely grateful for the thorough and friendly review  of our manuscript. I found these comments very important.  Please see the attachment. I hope that now the article has become better and clearer.
With best wishes and great appreciation.

Round 2

Reviewer 2 Report

I have checked the revised version of the article, and I agree with the author's changes. My decision is to accept it in the present form.

Good luck!